# Interplay between Gating and Block of Ligand-Gated Ion Channels

**DOI:** 10.3390/brainsci10120928

**Published:** 2020-12-01

**Authors:** Matthew B. Phillips, Aparna Nigam, Jon W. Johnson

**Affiliations:** 1Department of Neuroscience, University of Pittsburgh, Pittsburgh, PA 15260, USA; matt.phillips@pitt.edu (M.B.P.); apn20@pitt.edu (A.N.); 2Center for Neuroscience, University of Pittsburgh, Pittsburgh, PA 15260, USA

**Keywords:** ligand-gated ion channel, channel block, channel gating, nicotinic acetylcholine receptor, ionotropic glutamate receptor, AMPA receptor, kainate receptor, NMDA receptor

## Abstract

Drugs that inhibit ion channel function by binding in the channel and preventing current flow, known as channel blockers, can be used as powerful tools for analysis of channel properties. Channel blockers are used to probe both the sophisticated structure and basic biophysical properties of ion channels. Gating, the mechanism that controls the opening and closing of ion channels, can be profoundly influenced by channel blocking drugs. Channel block and gating are reciprocally connected; gating controls access of channel blockers to their binding sites, and channel-blocking drugs can have profound and diverse effects on the rates of gating transitions and on the stability of channel open and closed states. This review synthesizes knowledge of the inherent intertwining of block and gating of excitatory ligand-gated ion channels, with a focus on the utility of channel blockers as analytic probes of ionotropic glutamate receptor channel function.

## 1. Introduction

Neuronal information processing depends on the distribution and properties of the ion channels found in neuronal membranes. Channel gating, perhaps the most basic characteristic of ion channels, refers to the ability of ion channels to either open and allow transmembrane ion flux or to close and prevent ion flux. The gating mechanisms employed by ligand-gated ion channels are divided into three general categories, namely, activation, deactivation, and desensitization. Activation refers to the transition of ion channels from closed to open states following application of agonist. Deactivation refers to the transition of channels from open to closed states following removal of agonist. Desensitization is canonically defined as a decrease in the fraction of channels that are in the open state (termed open probability, or P_open_) in the maintained presence of agonist [1]. Desensitization is typically a direct consequence of agonist binding. A fourth gating mechanism that resembles desensitization but is not driven by agonist binding was referred to both as desensitization and inactivation [2,3,4], although inactivation is a term typically used to describe a different mechanism employed by voltage-gated channels [5]. Although driven by different underlying mechanisms, both desensitization and inactivation ultimately describe nonconducting channel states that do not respond to typical activating stimuli [1,5]. Gating mechanisms of ion channels are finely tuned and are essential to normal nervous system function, with even minor aberrations of channel gating often resulting in disease. While most known channelopathies involve dysfunction of voltage-gated channels, naturally occurring genetic variants that alter the gating of ligand-gated ion channels are increasingly associated with neurological disorders, including epilepsy, intellectual disability, and autism [6].

Studies of drugs that inhibit channel function provide valuable insight into ion channel gating mechanisms. Channel blockers, antagonists that bind in and prevent ion flux through ion channels, have been successfully used to probe both the structure of ion channel pores and the kinetics of channel gating. Channel gating requires conformational changes in or near the channel pore (i.e., the transmembrane ion conduction pathway), and channel blockers are known to interact differentially with channels in open, closed, inactivated, and desensitized states [3,7,8,9,10]. Thus, channel blockers are exceptionally well-positioned, both figuratively and literally, for use as analytic probes in studies of channel gating. Here, we discuss the interaction between channel block and channel gating of excitatory ligand-gated ion channels, with a focus on ionotropic glutamate receptors.

## 2. Reciprocal Interactions between Channel Block and Channel Gating

Channel gating can profoundly influence channel block, and channel block can profoundly influence channel gating. The initial binding of channel blockers often depends on gating state. Most blockers of ligand-gated ion channels can only enter and bind to the channel while agonist is bound and the channel is in the open state (Figure 1A). Such blockers are descriptively named open channel blockers and are the focus of this review. In some cases, open channel blockers are also termed “use-dependent” [11]. A blocker is termed use-dependent if inhibition by the blocker (1) requires activation of the channel, and (2) increases with duration of channel activation until an equilibrium between blocker binding and unbinding is reached (Figure 1A). The actions of almost all known ligand-gated ion channel blockers have been found to be at least partially dependent on channel opening, binding either exclusively or with much faster kinetics when the channel is open. 

Channel blocker unbinding also depends on gating transitions. If closure of the channel gate and agonist unbinding can occur while the blocker is bound, the blocker may become “trapped” in the channel (Figure 1C,D), unable to unbind until agonist is reapplied. Interestingly, some trapping blockers display the ability to escape from a fraction of blocked channels even after removal of agonist, a phenomenon termed “partial trapping” that is not fully understood [12,13,14,15,16]. On the other hand, sequential or “foot-in-door” channel blockers physically occlude closure of the channel gate (Figure 1E). The depth of the blocking site, size of the channel blocker, location of the channel gate, and gating-associated conformational changes all contribute to whether channels can close while the blocker is bound. These features dictate the structural interactions between channel blockers and the receptor’s gating machinery, which in turn determine the influence that the channel blocker can reciprocally exert on gating transitions. 

Bound channel blockers can affect gating transitions in three general ways. Blockers can: Alter agonist binding and/or unbinding kinetics;Stabilize channel open states;Stabilize channel closed states.

For example, the binding of large sequential blockers to open channels prevents both transition of channels into closed states (Figure 1E) and agonist unbinding [7,17,18,19,20]. In contrast, smaller trapping blockers can interact with either open or closed channel states and can therefore have many possible effects on channel gating (Figure 1D). For example, trapping blockers can stabilize open or closed channels and/or facilitate entry into or recovery from desensitized states [3,8,21]. The inherent intertwining of channel gating and block allows channel blockers to be leveraged as powerful tools for the study of ion channel structure and function.

## 3. Nicotinic Acetylcholine Receptors

Since the turn of the 20th century, the study of nicotinic acetylcholine receptors (nAChRs) has played a critical role in our understanding of ionotropic receptor biophysics and pharmacology. The very concept of transmitter receptors arose from the observation that application of nicotine to denervated striated muscle elicited muscle contraction, resulting in John Newport Langley’s inference of the presence of a “receptive substance” on muscle fibers [22]. The nAChR has served as the prototypical ion channel since its discovery, being the first ion channel to be isolated, characterized, structurally imaged, and cloned [23,24,25,26,27,28,29,30,31]. nAChRs additionally served as the subject for the first kinetic models of ion channel function [1,32] and for the development of patch-clamp electrophysiology [33,34]. Consistent with their vast historical importance, nAChRs mediate neuromuscular transmission and play key roles in nervous system function. Cholinergic signaling through nAChRs heavily modulates excitatory and inhibitory transmission in hippocampal and mesolimbic circuits, and thus is vital in shaping synaptic plasticity and learning [35,36,37,38,39,40]. Unsurprisingly, aberrant expression or activation of nAChRs is heavily implicated in many neurological and neuromuscular disorders, including addiction, schizophrenia, epilepsy, Alzheimer’s disease, myasthenia gravis, and Lambert–Eaton myasthenic syndrome, making nAChRs a major neurotherapeutic target [41,42,43,44,45,46,47,48,49]. 

nAChRs are excitatory, cation-selective members of the Cys-loop receptor superfamily, a major class of ligand-gated ion channels named for a conserved loop of 13 amino acid residues formed by disulfide-bonded cysteine residues in their extracellular domain. Other members of the Cys-loop family include the cation-selective serotonin (5-HT_3_) and zinc-activated (ZAC) receptors, as well as the anion-selective γ-aminobutyric acid (GABA_A_) and glycine (Gly) receptors [50]. Like all Cys-loop receptors, nAChRs are pentameric protein complexes with broad subunit diversity. nAChRs are assembled from a large catalog of subunits consisting of 10 α subunits (α1–10; although α8 is only expressed in avian species), four β subunits (β1–4), and the singular γ, δ, and ε subunits. This high subunit diversity is augmented further by the “sidedness” of each subunit, which allows the specific order in which subunits assemble to affect the receptor’s biophysical properties, resulting in more than 1000 possible nAChR subtypes [38]. All neuromuscular junction (NMJ) nAChRs are heteropentameric, composed of α1, β1, γ, and either δ or ε subunits at a respective ratio of 2:1:1:1. Neuronal nAChRs, on the other hand, assemble either as α-homomers or as heteromers composed of α(2–10) subunits complexed with β(2–4) subunits. Homomeric α7 and heteromeric α4β2 (2 α4, 3 β2) are the most commonly expressed nAChR subtypes in the brain [51,52,53]. 

All nAChR subunits possess a modular structure composed of a large extracellular N-terminal domain (NTD; the location of the Cys-loop), three-membrane spanning regions (M1-M3), a variable intracellular loop, another transmembrane region (M4), and a short extracellular C-terminal domain. Although most nAChR subtypes possess two acetylcholine (ACh) binding sites formed within the NTD at the interface between α subunits and their neighboring subunits, the precise location and properties of agonist binding sites differs broadly depending on subunit composition. For NMJ nAChRs, binding sites form at the α–γ or α–ε and at the α–δ subunit interfaces. Neuronal nAChRs again show greater diversity, with heteromeric receptors typically possessing two ACh binding sites located at α–β interfaces and homomeric receptors possessing five potential ACh binding sites [54,55,56]. The transmembrane regions (M1-M4) form a central, water-filled pore lined by the M2 transmembrane region that serves as a conduction pathway for the permeable cations Na^+^, K^+^, and Ca^2+^ [57,58]. Gating of nAChRs is initiated by binding of agonist in the extracellular domain, which begins a series of conformational changes that propagate to the transmembrane region and induce opening of the channel gate. Gating transitions of nAChRs are also heavily dependent on receptor subtype, but cycle through three basic states, namely, a resting closed state, an open state, and a desensitized state. nAChRs pass through multiple additional states after agonist binding prior to channel opening [59] and display many desensitized states. In fact, desensitization was first defined by Katz and Thesleff through the study of NMJ nAChR currents [1]. 

### Channel Block of nAChRs

As with nearly every aspect of ion channel research, nAChRs were also the first ion channels to be investigated using antagonists. Experiments utilizing the arrow poison curare and the snake venom α-bungarotoxin produced thorough descriptions of competitive antagonism and served as the starting point for the isolation of nAChRs and characterization of their function [29,60,61]. Early studies using channel blockers to investigate gating of ligand-gated ion channels also largely focused on nAChRs. Local anesthetics and barbiturates act as nonselective nAChR blockers (but also target voltage-gated Na^+^ channels [5] and GABA_A_ receptors [62], respectively) and were among the first drugs used to probe ligand-gated ion channel block. Treatment of muscle fibers with local anesthetics (most notably, the lidocaine derivatives QX-222 and QX-314) converted the normally single exponential decay of motor endplate currents into a double exponential decay [63,64], an effect also observed with the barbiturates thiopentone, amylobarbitone, and methohexitone [65,66]. These observations served as the basis of a slew of studies that characterized the basic features of ion channel block. Adams provided the first extensive characterization of use-dependent ligand-gated ion channel block, providing (1) compelling evidence that nAChR channel blockade was strongly voltage-dependent, suggesting that the blocker binding site was within the membrane electric field, and (2) the first model of sequential channel block of a ligand-gated ion channel (Figure 1E; [65,66,67]). Around the same time, Ruff proposed a similar conceptual model for sequential channel block of nAChRs [19]. This model was soon validated by Neher and Steinback, who used single-channel recordings to show that binding of QX-222 blocked current flow and that open, blocked channels could not close (Figure 1E; [20]). These pioneering studies laid the groundwork for the use of channel blockers as experimental probes for the study of ion channel gating and function. 

More recent studies of nAChR-channel blocker interactions provided further insight into nAChR gating mechanisms. Block of nAChRs by choline and millimolar concentrations of ACh prolong channel open times without affecting desensitization [10,68,69]. These findings led to the development of the “dual-gate” hypothesis of nAChR gating, which posits that nAChR activation and desensitization are mediated by distinct gates. Photolabeling experiments utilizing chlorpromazine further supported this dual-gate model, showing that (1) chlorpromazine has multiple binding sites in the nAChR channel and (2) the state of the receptor (i.e., desensitized, activated, or non-activated) directs the binding of chlorpromazine to specific sites in the channel [9,70,71,72]. Structural and functional evidence supporting the dual-gate model of activation and desensitization have been further found for an array of other pentameric ion channels [73], providing yet another example of the power of channel blockers as analytic tools.

## 4. Ionotropic Glutamate Receptors

Ionotropic glutamate receptors (iGluRs) are members of the pore loop superfamily of ion channels, integral membrane proteins that mediate the majority of ion flux across neuronal membranes [5]. Fast excitatory synaptic transmission in the central nervous system is primarily mediated by iGluRs, and proper functioning of iGluRs is vital to synaptogenesis, synaptic plasticity, signal integration, and information transfer [74,75]. Due to the integral roles iGluRs play in neuronal function and their ubiquitous expression, aberrant iGluR activity contributes to a wide variety of neuronal dysfunctions that can drive nervous system disorders [6,76,77,78,79,80,81,82].

iGluRs are divided into three main classes by structure: α-amino-3-hydroxyl-5-methyl-4-isoxazole-propionate receptors (AMPARs), kainate receptors (KARs), and *N*-methyl-D-aspartate receptors (NMDARs). A fourth division of the iGluR family, δ receptors, shares substantial sequence homology with other iGluR subtypes. Surprisingly, despite forming functional ion channels [83,84,85], δ receptors show no ligand-gated ion channel function [86,87,88,89] and are therefore not discussed in this review. All iGluRs assemble as complexes of four membrane-spanning subunits that form a central pore. Each iGluR subunit contributes exclusively to one subtype of iGluR: GluA1–4 form AMPARs, GluN1, GluN2A-D, and GluN3A-B form NMDARs, and GluK1–5 form KARs. Despite this wide diversity, all iGluR subunits possess a similar general structure (shown in Figure 2A using an NMDAR as an example) consisting of four discrete, semiautonomous domains, namely, an extracellular amino-terminal domain (ATD), an extracellular ligand-binding domain (LBD), a transmembrane domain (TMD), and an intracellular carboxy-terminal domain (CTD, which was deleted from the structure shown in Figure 2). Unlike nAChRs, each iGluR subunit possesses an agonist-binding site located within the LBD. The four TMDs of iGluRs form the pore, and thus the site of channel blocker binding (Figure 2B). Within the TMD lies the glutamine (Q)–arginine (R)–asparagine (N) (QRN) site, a site found at the tip of the re-entrant loop (M2 loop) in the iGluR pore (Figure 2C) that helps form the selectivity filter and plays a crucial role in the differential cation selectivity and channel block of the three iGluR classes [90,91,92,93]. Recent mid- and high-resolution structures of AMPAR [94,95,96,97] and NMDAR [21,98] TMDs provided great insight into iGluR gating transitions and channel block.

## 5. Characteristics of AMPA and Kainate Receptor Block

AMPARs and KARs display a number of common characteristics not shared with NMDARs. AMPARs and KARs can form both homomers and heteromers that possess four glutamate binding sites. Gating of AMPARs and KARs does not require all four subunits to be bound to an agonist, allowing single-channel currents to show multiple conductance levels depending on LBD occupancy [106,107,108]. Gating transitions of AMPARs and KARs are also very fast compared to NMDARs. Channel deactivation occurs within 5–10 ms of agonist removal and desensitization generates a >90% decrease in current within 20 ms of channel opening in the continued presence of an agonist [75]. These rapid gating transitions allow AMPARs and KARs to mediate the time course of the fast component of synaptic transmission [109].

### 5.1. Channel Block of AMPAR and KAR is Regulated by Channel Gating

Channel block of AMPARs and KARs is remarkably similar and is primarily governed by RNA editing at the QRN site in the channel pore. While the exons for all AMPAR and KAR subunits code for an uncharged glutamine (Q) at this site, RNA for the GluA2, GluK1, and GluK2 subunits can be edited, resulting in a change from the conserved glutamine to a positively charged arginine (R; [110,111,112]). GluA2-lacking AMPARs and GluK1/2-lacking KARs, which contain the unedited Q at the QRN site, are permeable to calcium (Ca^2+^) and are readily blocked by endogenous intracellular polyamines such as spermine [91,92,111,113,114,115,116]. However, incorporation of a single edited GluA2, GluK1, or GluK2 subunit into an AMPAR or KAR, i.e., a subunit with a positively charged R at the QRN site, abolishes Ca^2+^ permeability as well as polyamine block [113,114,115]. Interestingly, editing at the QRN site also controls inhibition of KARs by fatty acids, with only fully edited KARs displaying sensitivity to fatty acid inhibition [117]. Although fatty acid inhibition is only weakly voltage-dependent and may involve interactions with residues outside the pore [118], the involvement of the QRN site suggests a possible interaction within the pore. For the remainder of this section, we focus on AMPARs and KARs with Q at the QRN site of each subunit.

Polyamine block of iGluRs is strongly voltage-dependent and at least partially use-dependent [115,119,120]. Both AMPARs and KARs show birectifying responses in the presence of cytoplasmic polyamines [115], suggesting a common block mechanism. Recent cryo-electron microscopy (cryo-EM) structures of unedited GluA2(Q) receptors revealed the location and structure of the polyamine binding site in the pore [95,96]. Just below the QRN site lies a strongly electronegative portion of the channel, which likely contributes to both the cation selectivity of AMPARs [96] and the local membrane electric field [119,121,122]. Relief of polyamine block of AMPARs and KARs occurs via two separate mechanisms, i.e., permeation through the channel, which occurs at high positive voltages, and unblock to the cytoplasm, which occurs at negative voltages [115,123]. Polyamines are also trapping channel blockers and thus can only readily unbind when the channel is open [96,119]. Although trapping of polyamines seems counterintuitive, since they can unbind to either the intracellular or extracellular space, recent cryo-EM experiments provided compelling structural evidence of polyamine trapping. AMPAR structures produced by Twomey et al. suggested that closure of a gate near the extracellular entrance to the channel prevents polyamine permeation, while constriction of the selectivity filter prevents unbinding to the intracellular space by “pinching” the tail of the bound polyamine [96]. Thus, the gating state of AMPARs and KARs governs channel block by endogenous polyamines.

### 5.2. Effects of Auxiliary Proteins on Gating of AMPARs and KARs Modulates Block by Endogenous Polyamines

The interplay between gating and block of AMPARs and KARs is further regulated by the interaction of AMPARs and KARs with auxiliary proteins. Beginning with the discovery of the transmembrane AMPA receptor regulatory protein (TARP) stargazin [124], a wealth of studies identifying and characterizing additional auxiliary proteins have greatly increased our understanding of AMPAR and KAR regulation. Auxiliary proteins play an integral role in the trafficking, synaptic targeting, and gating of non-NMDAR iGluRs [121,122,125,126,127,128,129,130,131], and several in-depth reviews have been written concerning their function [131,132,133,134,135,136,137,138]. 

Through their regulation of channel gating, auxiliary proteins necessarily regulate channel block of AMPARs and KARs. Many auxiliary proteins, including stargazin, cornichon-3, and members of the CKAMP/Shisa family, substantially augment AMPAR currents by slowing desensitization and increasing mean channel conductance [122,139,140,141,142,143,144,145,146]. Similarly, the KAR auxiliary proteins Neto1 and Neto2 increase KAR currents by decreasing desensitization and increasing peak P_open_ ([121,129,130,131], but see [128]). Interaction with auxiliary proteins that increase channel conductance and/or P_open_ greatly attenuates polyamine block of AMPARs and KARs. Increasing P_open_ reduces the time that the channel spends in closed states, which in turn reduces polyamine trapping and facilitates permeation. Indeed, association of AMPARs with stargazin or cornichon-3 and KARs with Neto1/2 greatly increases polyamine permeation [121,122]. Although increasing P_open_ would also be expected to allow polyamines to more readily access their binding sites, AMPAR-auxiliary protein interactions were surprisingly found to slow the onset of polyamine block at negative voltages [147]. The enhancement of blocker permeation at positive voltages and reduction onset of block at negative voltages combine to profoundly weaken polyamine block. Association of AMPARs with stargazin or cornichon-3 attenuates polyamine block by 3–15-fold [122,140,146,147], and KAR association with Neto1/2 reduces block by 8- to 20-fold [121,148]. In contrast, the AMPAR auxiliary protein GSG1L reduces channel conductance while increasing polyamine-dependent rectification [125]. The structural underpinnings of this intrinsic relation between polyamine block and the regulation of AMPAR and KAR gating by auxiliary proteins are yet to be elucidated. Possible explanations include auxiliary protein-induced stabilization of receptor open states [146,149], alteration of pore structure due to interactions of auxiliary subunits with the AMPAR/KAR TMD [96,138,144], or interactions between auxiliary protein C-terminal domains and AMPAR/KAR intracellular domains [140,148].

### 5.3. Effects of Polyamine Block on Gating Transitions of AMPARs and KARs

Although much is known about how channel gating regulates AMPAR and KAR blockade by polyamines, there is a dearth of knowledge about how polyamine occupancy of the channel affects AMPAR or KAR gating. To our knowledge, there are no reports of modulation of KAR gating by polyamine block. Polyamine block of AMPARs is known to contribute to frequency-dependent synaptic facilitation, but this effect appears unrelated to polyamine effects on gating. Instead, at negative potentials, repetitive stimulation of AMPARs can cause an increase in the rate of polyamine unblock without a concomitant increase in the rate of block [119,120,150]. However, there is some evidence that polyamine binding may stabilize or accelerate entry into AMPAR closed states [119]. The presence of trapped spermine in the channel of unedited GluA2(Q) receptors causes a delay in channel activation upon presentation of agonist, suggesting that spermine stabilizes a closed receptor state. Furthermore, kinetic models suggest that the observed acceleration of channel deactivation in the presence of spermine could be explained by a doubling of the rate of channel closure [119]. Acceleration of channel closure could potentially result from polyamines emptying and excluding other permeant ions from the pore, an effect observed in studies of voltage-gated potassium channel block [151]. Stabilization of a channel closed state by polyamines could also be due to allosteric modulation, a mechanism suggested for certain blockers of Cys-loop receptors [73]. Given the roles Ca^2+^-permeable, non-NMDAR iGluRs play in both normal physiological and disease states [138,152,153,154,155], it is important to further our understanding of AMPAR and KAR block not only to better understand neuronal information processing, but also to aid in the design of more efficacious therapeutics.

## 6. Characteristics of NMDAR Channel Block

NMDARs display myriad biophysical properties unique amongst the iGluR family, including high Ca^2+^ permeability, slow gating kinetics, dependence on co-agonism for gating, and voltage-dependent block by magnesium (Mg^2+^) ions [75,156,157,158,159,160,161]. These characteristics allow NMDARs to control the magnitude and timing of Ca^2+^ influx during synaptic activity and therefore play a pivotal role in synaptic development and plasticity [162,163,164]. NMDARs are obligate heterotetramers, typically composed of two GluN1 subunits (eight splice variants), which bind glycine or d-serine, and two GluN2 subunits (GluN2A–GluN2D), which bind glutamate. A third group of subunits, GluN3A/B, also bind glycine/d-serine (although d-serine acts only as a partial agonist [165]) and can assemble with GluN1 and GluN2 subunits to form NMDARs activated by glutamate and glycine/d-serine. Interestingly, GluN3 subunits can also assemble just with GluN1 subunits to form unconventional NMDARs activated solely by glycine/d-serine. However, these GluN1/3 receptors only pass weak currents in physiological conditions, so their role in neuronal function is largely unknown [165]. Conventional NMDARs consisting of two GluN1 subunits and two GluN2 subunits rely on the binding of both glutamate and glycine/d-serine for activation [156,166] and, unlike AMPARs and KARs, require all four agonist binding domains to be occupied for the channel to transition to the open state [167,168,169]. Additionally, conventional NMDARs possess a conserved asparagine at the QRN site (Figure 2C) that confers high Ca^2+^ permeability even relative to Ca^2+^-permeable AMPAR and KARs, as well as sensitivity to block by Mg^2+^ [90,101,123]. 

Channel block of NMDARs has been extensively studied. Known, well-characterized NMDAR channel blockers with slow kinetics displayed some degree of use dependence. Another highly conserved key feature of NMDAR channel blockers is voltage dependence. Most NMDAR channel blockers are monovalent or divalent cations and display far greater inhibition at negative than at positive membrane potentials [12,13,15,17,170,171,172,173]. Due to their many roles in normal and pathological brain function, NMDARs are attractive targets for development of neurotherapeutics. NMDAR channel blockers are currently the most clinically useful NMDAR-targeting drugs and show great promise in the treatment of multiple nervous system disorders, including neurodegenerative diseases, major depressive disorder, and neuron death following ischemia [174,175,176,177,178,179,180,181,182]. 

NMDAR channel blockers display a strikingly diverse array of clinical effects, despite sharing overlapping binding sites and a similar general mechanism of inhibition ([99,183]; the putative blocking site for memantine is shown in Figure 2B,C). For example, the clinically relevant blockers memantine and ketamine share similar chemical properties (Table 1) and binding kinetics but possess vastly different effects on brain function. Ketamine is a drug of abuse and poorly tolerated, but possesses impressive efficacy in treating neuropathic pain and major depressive disorder [176,180,184,185]. On the other hand, memantine possesses weaker efficacy in treatment of neuropathic pain and little to no effect on major depressive disorder, but is well-tolerated with few side effects and shows efficacy in the treatment of neurodegenerative disorders such as Alzheimer’s disease [177,179,186,187,188,189,190]. The striking diversity in the clinical effects of NMDAR channel blockers may in part arises from their diverse effects on channel gating. Nearly all known NMDAR channel blockers show some effect on channel gating [191] and channel blockers are found to modulate nearly every aspect of gating [3,7,8,12,13,17,173,192,193,194,195,196,197,198,199,200]. 

### 6.1. Sequential Blockers of NMDARs Prevent Channel Closure and Agonist Dissociation

The sequential/foot-in-door blockers 9-aminoacridine (Table 1), tetrapentylammonium, and the amantadine derivative IEM-1857 (synthesized at the Institute of Experimental Medicine (IEM), St. Petersburg, Russia) are thought to force NMDARs to remain in open states by sterically prohibiting gate closure after entering the channel [7,17,173,195,201]. Importantly, occupancy of the channel by IEM-1857, tetrapentylammonium, or 9-aminoacridine also prevents agonist dissociation and channel desensitization [7,17,173], suggesting that blocker unbinding and subsequent channel closure are required for agonist dissociation. This finding is consistent with models of sequential channel block of nAChRs proposed by [19,20,66,67]. An experimental procedure used to test whether a channel blocker prevents channel closure and agonist dissociation is to determine if the blocker induces “tail currents”. A tail current is a transient increase in receptor-mediated current observed upon rapid and simultaneous removal of blocker and agonist from the extracellular solution. If a blocker prevents channel closure, channels pass through the open, unblocked state following blocker unbinding, resulting in a tail current. However, any antagonist that unbinds more quickly than agonists can induce tail currents; thus, observation of tail currents does not provide unambiguous evidence that a blocker acts via a sequential mechanism. More powerful evidence that a blocker prevents channel closure can be provided by (a) observation that a blocker chops single-channel currents into “bursts” of brief openings, and that the total channel open time during bursts is independent of blocker concentration [20], and (b) observation that the blocker concentration that inhibits responses by 50% (the IC_50_) is inversely proportional to the receptor’s P_open_, a prediction that can be tested, e.g., by recording the IC_50_ of a blocker over a range of agonist concentrations [191]. The finding that channel occupation by sequential blockers prevents agonist unbinding as well as channel closing provided fundamental information on state transitions of ligand-gated ion channels. 

**Table 1 brainsci-10-00928-t001:** NMDAR channel blockers and their effects on gating.

Compound	Structure	Type of Blocker	Effects on Gating
Magnesium	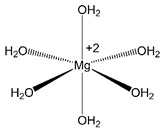	Unclear—due to fast unblocking kinetics, trapping of Mg^2+^ has not been directly demonstrated.	None [13,202].
9-aminoacridine	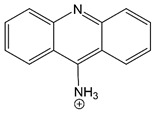	Sequential [7,201].	Stabilizes open state [7,201]. Prevents agonist dissociation [7,201].
IEM-1754	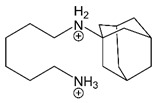	Depolarized potentials: sequential [173]. Strongly negative potentials: trapping [173].	Depolarized potentials: Stabilizes open state [173].
Amantadine	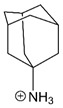	Partial trapping [12,13].	Accelerates channel closure of native NMDARs and GluN1/2B receptors [8].
Memantine	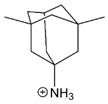	Partial trapping [8,16,197,203,204].	Slows GluN1/2A receptor recovery from Ca^2+^-dependent desensitization [3].
Ketamine	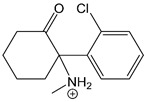	Trapping [204].	Accelerates GluN1/2B receptor recovery from desensitization [3].

Magnesium is depicted coordinating six water molecules, and all organic blockers are depicted in bond-line format. Blockers structures are scaled to depict approximate relative sizes.

Organic channel blocking compounds were remarkably useful in determining the location of the channel gate itself. The size of a blocking molecule is a key determinant of whether the blocker prevents channel closure or is trapped in the channel upon gate closure. Experiments comparing block by IEM-1857 and the similar but smaller blocker IEM-1754 (Table 1) found that while binding of IEM-1857 prevented channel closure independent of voltage, IEM-1754 only prevented channel closure at relatively depolarized membrane potentials. At more hyperpolarized potentials, IEM-1754 is “pulled” by the membrane electric field deeper into the channel where it no longer prevents channel closure, instead acting as a trapping channel blocker [173]. The voltage dependence of IEM-1754 block, as well as its interactions with permeant ions, demonstrated that IEM-1754 has two blocking modes, one that associates with a shallower site and places the bulk of the molecule in the way of the gate, and a second that associates with a deeper site and permits closure of the gate [173,205,206]. This finding strongly supported the idea that the NMDAR channel gate lies at the extracellular entrance to the channel, an idea that was recently validated by crystal and cryo-EM structures of ligand-bound NMDARs [98,207].

### 6.2. Trapping Channel Blockers Modulate NMDAR State Transitions

Trapping channel blockers display more subtle effects on gating than sequential blockers. Early studies using a combination of patch-clamp electrophysiology and kinetic modeling concluded that the amino-adamantane derivatives memantine and amantadine and the phencyclidine derivative NEFA have clear effects on channel gating [12,192,193,197]. Initial proposals for the effects of amino-adamantane derivatives on NMDAR gating were wide-ranging, including models that suggested memantine and amantadine could stabilize open receptor states, as well as models that suggested memantine may stabilize closed receptor states [12,192,197]. It is possible that these discrepancies arose from the abilities of amino-adamantane derivatives to escape from some blocked channels after agonist removal (partial trapping) and to inhibit NMDARs via association with a site accessible in the absence of agonist [12,16,192,197,208,209]. 

Thorough evidence that amino-adamantane derivatives affect closed-state transitions came through investigation of the discrepancy between the equilibrium dissociation constant (K_d_) and potency (represented by IC_50_) of amantadine. The relation between K_d_ and IC_50_ depends directly on how a channel blocker affects channel transitions after binding. K_d_ < IC_50_ implies that a blocker stabilizes channel open states. This is the case for sequential blockers, which inhibit less effectively as P_open_ decreases (IC_50_ = K_d_/P_open_, see Section 6.1; [5,191]). In contrast, K_d_ > IC_50_ implies that a blocker’s mechanism of inhibition likely involves stabilization of channel closed states, either through decreasing the rate of channel opening, increasing the rate of channel closure, or both. Such blockers therefore have two inhibitory actions: (1) blocking current flow through open channels and (2) stabilization of closed channels. Amantadine is an example of such a dual-mechanism channel inhibitor. Amantadine’s K_d_ (110 μM) is considerably greater than its IC_50_ (~35 μM; [8,15,192,210]). Investigation of amantadine block of single-channel and whole-cell NMDAR current revealed that binding of amantadine not only accelerates channel closure, but that this acceleration of channel closure is actually the predominant mechanism of inhibition by amantadine at concentrations lower than 100 μM [8].

Recent studies reported additional drug-specific and NMDAR subtype-specific effects of channel blockers on gating transitions. Investigation of mechanisms by which memantine and ketamine preferentially target distinct populations of NMDARs led to the discovery that memantine and ketamine have differential, subtype-specific effects on NMDAR desensitization [3]. While ketamine accelerated recovery from desensitization of GluN1/2B receptors, memantine binding profoundly slowed recovery from desensitization of GluN1/2A receptors. The effect of memantine on GluN1/2A receptor desensitization was not observed in low-Ca^2+^ conditions, suggesting that memantine stabilizes a Ca^2+^-dependent desensitized state of GluN1/2A receptors. A comparison of IC_50_ values measured in low and high Ca^2+^ conditions with K_d_ values predicted by a kinetic model found that in high Ca^2+^, K_d_ > IC_50_, whereas in low Ca^2+^, K_d_ ≈ IC_50_, suggesting that memantine only alters GluN1/2A gating when Ca^2+^-dependent desensitization can occur [3]. 

Visualization of NMDARs bound to trapping channel blockers was provided by recent structural studies. Song et al. crystalized the closed GluN1/2B channel in complex with the high affinity blocker MK-801 and utilized long-timescale molecular dynamics to investigate the mechanism of block by MK-801 and memantine [21]. Both blockers were found to bind within the central cavity of the ion channel and promote closure of the channel gate [21], perhaps via a mechanism similar to amantadine [8]. Although this result may seem to contrast with the previous finding that memantine did not affect GluN1/2B receptor desensitization [3], it is important to note that (1) memantine could affect GluN1/2B channel closure without affecting desensitization, and (2) the crystalized MK-801-NMDAR construct lacked both the ATD and CTD, which play key roles in gating and desensitization [98,211,212,213,214,215]. Stabilization of closed channels by NMDAR channel blockers could have profound physiological implications by effectively increasing the potency of blockers under certain conditions. For example, the ability of memantine to stabilize a Ca^2+^-dependent desensitized state suggests a logical mechanism for neuroprotection: preferential inhibition of NMDARs in cellular populations subjected to pathological levels of Ca^2+^ influx, i.e., NMDARs likely to mediate excitotoxic cell death [79,216,217,218]. 

### 6.3. Channel Block by Mg^2+^ Does Not Appear to Affect NMDAR State Transitions

The majority of NMDAR channel blockers affect gating, but at least one blocker exists as an exception to this rule: Mg^2+^ (Table 1). Binding of Mg^2+^ to the NMDAR channel does not prevent gate closure, agonist dissociation, or desensitization [7,13,157,219]. Mg^2+^ boasts nearly equivalent K_d_ and IC_50_ values [202], further suggesting that Mg^2+^ occupancy of the channel has no effect on state transitions. The unusual ability of Mg^2+^ to block without altering gating could be due to its small size. A large conformational change in the extracellular region of the NMDAR channel is associated with gating, a conclusion supported by structural studies [98] and the observation that large organic blockers prevent channel closure. Although smaller organic blockers generally permit channel closure, stabilizing or destabilizing interactions with channel residues may alter channel gating. It is possible that the small size of Mg^2+^ (which is likely to be mostly dehydrated when blocking the channel [103]), coupled with its limited interactions with channel residues outside of the ion selectivity filter [103], allows binding in the NMDAR channel without affecting gating machinery. Also, in contrast to most organic blockers, Mg^2+^ has not been directly shown to act as a use-dependent open channel blocker or as a trapping blocker. Mg^2+^ displays extremely rapid binding and unbinding kinetics [157,219], preventing accurate determination in whole-cell recordings of the rapid component of block or unblock, measurements required for demonstration of use dependence and trapping. Kinetic modeling studies, however, suggested that Mg^2+^ does indeed act as an open channel blocker [13]. 

Despite the lack of effects of Mg^2+^ block on NMDAR gating, depolarization-induced Mg^2+^ unblock clearly depends on gating. Mg^2+^ unblock from GluN1/2A and GluN2B receptors displays a slow component as well as an extremely rapid component [220,221,222,223,224,225]. Although kinetic models in which Mg^2+^ block affects gating transitions and/or agonist binding rates reproduced slow Mg^2+^ unblock [222,224], substantial experimental evidence demonstrated that Mg^2+^ does not affect NMDAR state transitions [7,13,157,219]. This disagreement was reconciled by the discovery of the inherent (i.e., Mg^2+^-independent) voltage-sensitivity of NMDAR gating, which underlies the slow component of Mg^2+^ unblock [221,225]. Thus, the interplay between Mg^2+^ block and NMDAR gating is unidirectional, whereby Mg^2+^ block depends on NMDAR gating, but NMDAR gating is unaffected by Mg^2+^ block. 

## 7. Conclusions

Channel blockers are invaluable tools for the study of channel gating. The diverse array of effects that blockers exert on gating have facilitated numerous seminal discoveries into both the structure and the function of receptor channel gating machinery. The abilities to alter the rate of gating transitions and stabilize/destabilize channel states enable blockers to act as dual-mechanism drugs, both inhibiting current flow and modulating receptor function. The stabilization of specific receptor states also may contribute to the surprising diversity in the clinical effects of channel blockers. Future research determining the structural mechanisms by which channel blockers influence gating may aid in the directed design of more clinically efficacious neurotherapeutics.

## Figures and Tables

**Figure 1 brainsci-10-00928-f001:**
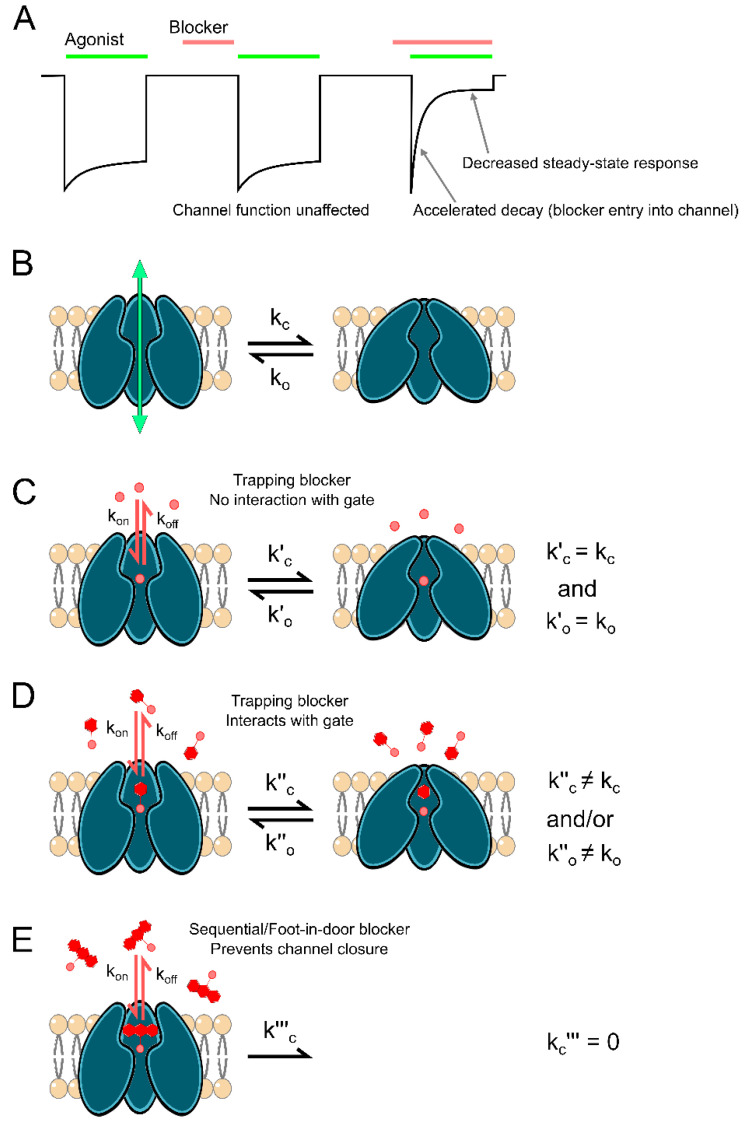
Interplay between channel gating and open channel block. (**A**) Schematic depicting inhibition of current (black line) by a prototypical open channel blocker. Three agonist applications (green bars) are shown. The first agonist application in the absence of blocker shows the control response. The second agonist application, which follows the application and removal of a blocker (red bar), shows that the blocker cannot access its binding site when the channel is closed. The third agonist application, which is made in the presence of a blocker, shows that the blocker can access its binding site and inhibit agonist-activated current when the channel is in the open state. Entry of a blocker into open channels accelerates the apparent decay of the response and decreases the steady state response. Because the blocker cannot bind until the channel opens, peak current in response to the first agonist application in the presence of the blocker may be unaffected, as shown here. However, if blocker binding is fast relative to current activation kinetics, the peak response may be reduced. (**B**) Ion channels can transition between open, ion permeable states and closed, impermeable states. k_c_ is transition rate into closed state and k_o_ is transition rate into open state. (**C**–**E**) The size of channel blocking compounds (red) and depth of the blocking site affects blocker interactions with the channel gate. (**C**) Small channel blockers, such as inorganic cations, can block open channels without preventing channel closure or affecting gating transitions. k_on_ is blocker binding rate and k_off_ is blocker unbinding rate. When the channel is blocked by a blocker that does not interact with the gate, channel closing rate is k′_c_ and channel opening rate is k′_o_. (**D**) Small-to-intermediate-sized organic channel blockers can block open channels without preventing channel closure, but nevertheless can interact with the channel gate, either accelerating or decelerating gating transitions. When the channel is blocked by a blocker that interacts with the gate, channel closing rate is k″_c_ and channel opening rate is k″_o_. (**E**) Large, organic, sequential/foot-in-door blockers can block open channels and prevent channel closure. k‴_c_ is channel closing rate when the channel is blocked by a sequential/foot-in-the-door blocker.

**Figure 2 brainsci-10-00928-f002:**
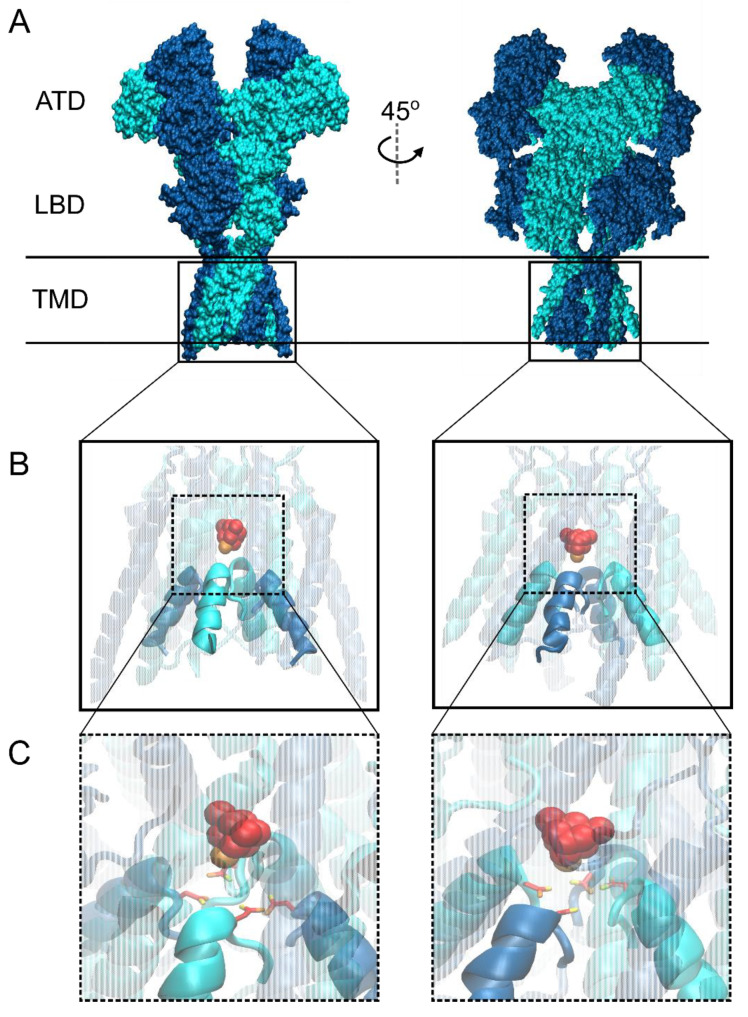
General N-methyl-D-aspartate (NMDA) receptor structure and putative blocking site. (**A**) Recently published structure of N-methyl-D-aspartate receptor (NMDAR) in an “active” state showing domain topology shared by all iGluR subtypes (ATD, amino-terminal domain; LBD, ligand-binding domain; TMD, transmembrane domain; Protein Data Bank (PDB) code 6WHT; [98]). GluN1 subunit is depicted in dark blue and GluN2B in cyan. Horizontal lines show the approximate locations of the outer and inner surfaces of the membrane. (**B**) Blow-up of NMDAR TMD (boxes in A) with docked channel blocker memantine (space-filled; carbons are red, nitrogen is orange) displaying typical site of channel block. Most channel blocking compounds show intimate interaction with the external tip of the iGluR selectivity filter formed by the re-entrant M2 loops of each subunit (opaque; M1, M3, and M4 transmembrane helices are transparent for visualization of blocking site). (**C**) Magnified view of memantine coordination by the QRN site asparagine residues GluN1 N616 and GluN2B N615, which are critically involved in NMDAR channel blocker binding [99,100,101,102,103]. Autodock Vina was used for molecular docking of memantine to PDB 6WHT, and structural images were prepared using the program Visual Molecular Dynamics (VMD) [98,104,105].

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
