# Peer review of "Interplay between Gating and Block of Ligand-Gated Ion Channels"

_brainsci, 2020, doi:10.3390/brainsci10120928_

Round 1
Reviewer 1 Report
The review is very interesting. It summarizes the major findings in the ligand-gated ion channels physiology field, and the use of different compounds as a channel blockers to study the gating mechanisms of these channels.
Major comments:
- The introduction has many technical and language impressions regarding ion channel mechanism, and gating properties that can be misleading to the reader, as examples in topics such as:
Desensitization/Inactivation (not the same thing)
Gating mechanism description (see text)
- Many of the statements in the introduction need to be associated with a strong reference, ideally a review. The authors only have two references in the entire introduction
- There are quite complex concept that needs to be clarified (or unpacked from one sentence) in the introduction that will be helpful later in the text
- The mention of voltage-gated ion channel in the introduction is misleading the main focus is on ligand-gated ion channels. Please remove it
- In the section "Reciprocal interactions between channel block and channel gating". Is has to start to a clear explanation why some blockers are state-dependent and also provide clear examples. The first paragraph is very confusing and need to be rephrase in more comprehensive way. I would suggest to add to Figure 1; 1) A cartoon to illustrate the differences between a strong use-dependent blocker vs. weak one, and how the traces look like, and 2) examples of current traces in the presence of the different blockers described there. How those changes in the kinetics are reflected in the recordings.

Reviewer 2 Report
This is a nicely written and comprehensive review talking about the effects that blockers of ligand-gated (LGIC) ion channels exert on the gating properties of these channels with special emphasis on ionotropic glutamate receptors.
I have only to two minor critical points to be addressed:
- In a chapter on "Channel block of nAchRs" authors talk about experiments studying the effect of lidocaine derivatives on the ion channel block in a context of their action on acetylcholine receptor in neuromascular preparations for measuring end plate potentials. Those are one of the pioneer experimental works in the field that are definitely critical to mention in the review. Yet, I think, it would be nice to give a remark for the readers that lidocaine and its derivatives can be considered as non-selective nAchR blockers since they are well-known blocker of voltage-gated sodium channels. Similar remark should be given for the barbiturates that are broad LGIC modulators with higher affinity to GABA-A receptor. May be it would be worth to give additional example how selective, modern, blockers of nAchR affect its gating properties.
- At the end of a chapter on "Characteristics of NMDAR channel block" authors talk about mechanisms underlying NMDA receptor block by Mg2+ ions. I think it would be critical to add some additional information stressing on the phenomena of fast and slow Mg2+ unblock for this paragraph.
Reviewer 3 Report
Authors of this manuscript provide a detailed review on the blockers that act on ligand-gated ionic receptors, such as nAChRs, NMDAR, KAR, and AMPAR. The major comment is that the manuscript is not concisely and clearly written and is hard to quickly grab useful information and knowing the concepts from the text. My comments are below.
- The text appears to only convey one message, that is, various types of blockers can affect the gating of their corresponding ion channels, and vice versa. It would be better to provide more mechanical information about how the blockers' inhibitory effects can be affected by the states of ion channels(open/close/inactive/deactive states), what is the structural basis for the observed blocking effects? e.g. what are the residues in the pore region and/or ligand binding regions that may impact ligand binding. In addition of discussing the ephys recording observations of different blockers of the different types of receptors, more in-depth views on gating and block of ion channels would spark more interest to broader readers.
- It could make the content clearer by providing more graphical illustrations or more tables, to summarize the blockers for nAChRs, AMPAR, KARs, as well.
- Fig2. putative blocking sites should be highlighted, e.g. interacting residues in the receptor. Otherwise the figure provides no information. The abbreviations, ACD, LBD, TMD, could also be explained in the figure legend.
- It would be better to also provide structures of nAChR2 bound with/without blockers and discuss how the binding of antagonists can block the channel and impact the gating, and the structural basis for the interaction between gating and block.
Round 2
Reviewer 1 Report
The authors have addressed all the comments/ suggestions satisfactorily
Reviewer 3 Report
Authors have answered my comments. Nice job.